# Parp Inhibitors and Radiotherapy: A New Combination for Prostate Cancer (Systematic Review)

**DOI:** 10.3390/ijms241612978

**Published:** 2023-08-19

**Authors:** Inés Rivero Belenchón, Carmen Belen Congregado Ruiz, Carmen Saez, Ignacio Osman García, Rafael Antonio Medina López

**Affiliations:** 1Urology and Nephrology Department, University Hospital Virgen del Rocío, 41013 Seville, Spain; nachosman79@hotmail.com (I.O.G.); rantonio.medina.sspa@juntadeandalucia.es (R.A.M.L.); 2Biomedical Institute of Seville (IBIS), 41013 Seville, Spain; csaez1@us.es

**Keywords:** PARP inhibitors, radiotherapy, prostate cancer

## Abstract

PARPi, in combination with ionizing radiation, has demonstrated the ability to enhance cellular radiosensitivity in different tumors. The rationale is that the exposure to radiation leads to both physical and biochemical damage to DNA, prompting cells to initiate three primary mechanisms for DNA repair. Two double-stranded DNA breaks (DSB) repair pathways: (1) non-homologous end-joining (NHEJ) and (2) homologous recombination (HR); and (3) a single-stranded DNA break (SSB) repair pathway (base excision repair, BER). In this scenario, PARPi can serve as radiosensitizers by leveraging the BER pathway. This mechanism heightens the likelihood of replication forks collapsing, consequently leading to the formation of persistent DSBs. Together, the combination of PARPi and radiotherapy is a potent oncological strategy. This combination has proven its efficacy in different tumors. However, in prostate cancer, there are only preclinical studies to support it and, recently, an ongoing clinical trial. The objective of this paper is to perform a review of the current evidence regarding the use of PARPi and radiotherapy (RT) in PCa and to give future insight on this topic.

## 1. Introduction

Prostate cancer (PCa) is the most common tumor diagnosed in men worldwide, ranking as the third most common cause of cancer death in Europe and the second in the United States [1]. However, PCa is a complex and heterogeneous condition with different degrees of aggressiveness, ranging from indolent to lethal forms [2]. This diversity of PCa presentations and stages calls for the availability of a broad spectrum of treatment options that vary from active surveillance to surgical procedures, radiation therapies, and intensive multimodal and systemic approaches [3].

This scenario has made it necessary to search for new therapeutic strategies and new combination treatments. One of the most promising strategies involves the DNA-damage response (DDR) pathway. DDR gene alteration creates a reliance on poly(adenosine diphosphate-ribose) polymerase (PARP)-1 for repairing DNA, which causes cancer cell death when PARP-1 is blocked [4]. PARP inhibitors are a new class of targeted drugs developed recently, which offer a novel approach to treating PCa by utilizing mutations in germline and somatic DNA damage repair (DDR) pathways, which allows for a genetically stratified treatment strategy [5]. This phenomenon called “synthetic lethality” is based on the theory that two different molecular pathways, which do not cause cell death when disrupted individually, can result in cell death when inhibited at the same time [6].

Remarkably, PARPi in combination with ionizing radiation, has demonstrated the ability to enhance cellular radiosensitivity in different tumors [7,8,9,10,11,12]. It is known that all kinds of radiation have an effect on the exposed biological systems that may be positive or negative [13]. Radiotherapy uses ionizing radiation (IR) from low linear energy transfer (LET) X-rays (photons) to treat tumors. However, this IR can cause acute and long-term adverse events due to the irradiation of surrounding tissues [14]. The aim of the radiation therapy is to kill the tumors cells while preserving the nearby healthy ones [15]. This objective is not always easy to achieve, as there is a permanent interplay between ionizing radiation (IR) and biological/cellular elements that can transpire in two primary ways: direct ionization or excitation of large molecules like DNA, or more commonly, indirect initiation via the breakdown of water into reactive oxygen species [16]. Among these, hydroxyl radicals are particularly prominent, as they can later engage with neighboring large molecules. The critical cellular target that drives tumor cell killing is DNA [17]. Indeed, for low-LET radiation, 1 Gy dose produces approximately 1000 DNA single strand breaks (SSBs), 40 DNA double strand breaks (DSBs), and 1300 DNA base lesions [18]. However, this DNA damage caused by the IR does not always lead tumor cells to die, as DNA has a sophisticated signaling network that is able to detect DNA damage and consequently initiate a complex repair process [19]. It is precisely here that PARPi play their key role by blocking the DNA repair mechanism and therefore maintaining DNA damage, which finally drives tumor cells to death [20].

The rationale is that exposure to radiation leads to both physical and biochemical damage to DNA, prompting cells to initiate three primary mechanisms for DNA repair. Two of the repair pathways are double-stranded DNA breaks (DSB): non-homologous end-joining (NHEJ) and homologous recombination (HR). The third one is a base excision repair (BER), which is a single-stranded DNA break (SSB) [21] that occurs more frequently in the context of external beam radiotherapy and is the only possible mechanism of repair in BRCA-mutated cells. SSBs are the most common DNA lesions and are relatively easily repaired, while DSBs represent a higher threat to genome integrity as they are far more difficult to repair [22,23]. However, sometimes SSBs cannot be adequately repaired and are converted to DSBs, which are highly mutagenic and cytotoxic when left unrepaired, interfering with important cellular processes and survival [23]. Regarding DSBs, the NHEJ pathway is responsible for mending the majority of lesions that have two ends. However, when DNA replication forks collapse in the S phase and create DSBs with only one end, NHEJ becomes hazardous due to its potential to create chromosomal rearrangements by reconnecting DNA ends from distinct chromosomes. Consequently, NHEJ (also known as error-prone) is deliberately restrained at replication forks through elements of the secondary major DSB repair pathway, HR [23]. In that sense, PARP1 contributes to the HR pathway of DSB repair by promoting rapid recruitment of MRE11, EXO1, BRCA1, and BRCA2 to DNA damage sites [24,25,26,27]. Additionally, PARP1 counters NHEJ (the alternative pathway for DSB repair) by inhibiting the attachment of the NHEJ protein Ku to the ends of DNA, which initiates the NHEJ repair mechanism [28,29,30,31]. Regarding SSB repair, PARP1 aids in the recruitment of the scaffold protein XRCC1 to the sites of DNA damage to repair it in a mechanism known as BER [32,33]. Together, this explains why PARP inhibitors, acting against the DDR pathway, enhance the radiotherapy effects that provoke DNA damage [34]

Furthermore, PARP inhibitors destabilize replication forks via PARP DNA entrapment and induce cell death via replication stress-induced mitotic catastrophe [35]. PARP1 interacts with DNA replication machinery during S phase, and in response to replication stress, that leads to uncoupling between DNA polymerase and helicase activities, which generates single-stranded DNA (ssDNA) [36,37,38,39]. When this occur, RPA binds ssDNA and recruits the S/G2 checkpoint kinase ATR to induce cell cycle arrest [40]. Thus, replication checkpoints prevent accumulation of ssDNA and exhaustion of RPA and thereby safeguard against fork breakage [41]. In response to replication stress, PARP1 decelerates the progression of replication forks to facilitate the reversal of forks by counteracting the RECQ1 helicase [42,43,44]. It safeguards replication forks against deterioration caused by the MRE11 nuclease [45], reinforces the stability of RAD51 nucleofilaments at paused forks in conjunction with PARP2 [46], and triggers the activation of the S-phase checkpoint kinase CHK1 [47]. Finally, PARP1 also regulates replication and DNA repair at the transcription level by stimulating the activity of the transcription factor E2F1, which regulates the expression of replication and HR genes [48].

In this scenario, PARPi can serve as radiosensitizers, driving tumor cells to death, blocking the reparation of damaged DNA caused by radiotherapy by leveraging the BER pathway, heightening the likelihood of replication forks collapsing that leads to the formation of persistent DSBs and inhibiting the HR and NHEJ repair pathways [49].

The objective of this paper is to perform a systematic review of the current evidence regarding the use of PARPi and radiotherapy (RT) in PCa and to give future insight on this topic.

## 2. Methods

### 2.1. Search Strategy

In May 2023, we conducted a systematic literature search through PubMed, Scopus, and Web of Science databases using the PICO criteria [50]:

P (Population): Prostate cancer cells, xenografts, or patients;

I (Intervention): Combinations of PARPi and radiotherapy;

C (Comparator): No comparator was mandatory;

O (Outcomes): Safety and oncological outcomes.

We utilized a specific search strategy to gather relevant data and evaluated the quality of the studies using a standardized methodology.

### 2.2. Article Selection

We followed the Preferred Reporting Items for Systematic Reviews and Meta-Analyses (PRISMA) guidelines [51,52]. Two authors (I.R.B. and B.C.R.) independently screened the articles based on our inclusion and exclusion criteria, with disagreements resolved by a third author (I.O.G.).

By following this method, we identified a large number of articles that afterwards went through a selection process (Figure 1).

Study identification: Using the above-explained search strategy, we found 77 articles regarding the combination of PARPi and radiotherapy (RT) in prostate cancer. We identified 21 reviews, systematic reviews, and meta-analyses; 3 clinical trials (none of them randomized); and 53 original articles. Most of the original articles were preclinical studies.

Screening: After duplicates were removed, 75 articles were screened by title and abstract.

Eligibility: 18 records were assessed via screening of the full text. The inclusions were: (a) reviews, systematic reviews, meta-analyses, clinical trials, and original articles; (b) the combination of radiotherapy and PARPi for prostate cancer treatment. Among the exclusion criteria we defined were: (a) Non-English/Spanish texts; (b) editorials, comments, and letters; (c) non-prostate cancer tumors and; (d) drug and molecular radiotherapy.

Study analysis: Finally, seven studies were selected for eligibility for the study analysis.

## 3. Results

### 3.1. PARP Inhibitors in Prostate Cancer

The two main repair mechanisms for double-stranded breaks (DSBs) are homologous recombination (HR) and non-homologous end joining repair (NHEJ) [53]. The signaling pathway of the HR system is executed by the sequential recruitment of repair proteins into the chromatin surrounding the lesion. The first sensor of the DSBs is the MRN complex (MRE11-RAD50-NBS1), which is attached to both sides of the breaks to signal them. Subsequently, recruitment and accumulation of regulated proteins occur by a complex mechanism that employs phosphorylation and ubiquitination mediated by various kinases, including BRCA1 and CtIP [54]. Thus, maintenance of the repair machinery is critical to protecting cells from DNA damage and preventing tumor processes [55].

Poly(ADP-ribose) (PAR) is involved in different cellular processes such as DNA replication, transcription, repair, and cell death. PARPs are enzymes implicated in PAR synthesis. PARP-1 is a crucial sensor protein for DNA damage that activates signaling pathways that promote appropriate cellular responses and exhibits significantly increased catalytic activity leading to the induction of poly ADP-ribosylation (PARylation) [56]. PARylation is a process that involves breaking down nicotinamide adenine dinucleotide (NAD+) and transferring the resulting ADP-ribose to either PARP-1 itself (autoPARylation) or other specific proteins (PARylation). These activities trigger PARP-1 and other DNA repair enzymes to start DNA repair processes by modifying the structure of chromatin and directing DNA repair factors to the site of damage [57,58]. Both PARP-1 and PARP-2 facilitate the recruitment and activation of BER factors and consequently facilitate DNA single-strand break (SSB) repair. Moreover, PARP-1 participates in repairing DNA DSBs (through NHEJ and HR) and correcting DNA replication errors [59].

On the other hand, BRCA2 mutations are recognized as significant risk factors for developing PCa. Homologous recombination repair (HRR) is a repair mechanism that depends on the BRCA1/2 genes. Consequently, tumor cells with deficient BRCA1/2 genes are unable to repair DNA damage through HRR and rely on PARP proteins for the restoration of single-strand breaks (SSBs). When PARP proteins are inhibited by PARPi, DNA repair cannot occur, leading to subsequent tumor cell death [60].

Given this background, the utilization of PARPi in PCa is supported by two key factors: the elevated occurrence of genetic mutations in PCa and the synthetic lethality concept. Genetic mutations in PCa involve both germline and somatic alterations. Germline mutations impact all cells within the body and can provide valuable insights for genetic counseling. Somatic alterations, on the other hand, are exclusive to tumor cells and arise as a consequence of inherent genome instability within the tumor itself, as well as clonal selection triggered by prior treatments [61]. In PCa, PARPi act through two mechanisms: (1) competitively binding to the active site, thereby preventing the repair of SSBs and favoring their conversion into DSBs [59] and; (2) trapping PARP-1 onto the damaged DNA, inhibiting autoPARylation [58]. Additionally, PARP-1 plays a role in delaying the progression of replication forks, which further impedes the repair of DSBs, ultimately leading to cell death [60,62]. Together, they contribute to the accumulation of DNA double-strand breaks (DSBs) that Homologous recombination repair (HRR)-deficient cells cannot repair efficiently (Figure 2).

Olaparib, rucaparib, niraparib, and talazoparib are PARPi with different mechanisms of action and distinctive trapping capacities that have been proven in PCa (Table 1). Talazoparib is the one with the most trapping capacity, and rucaparib is the one with the least capacity.

PROfound is a phase III clinical trial (CT) investigating Olaparib in patients with metastatic castration-resistant prostate cancer (mCRPC) and HRR gene mutations progressing after enzalutamide or abiraterone acetate plus prednisone (AA) (second-line setting). This CT demonstrated that patients treated with olaparib had significantly longer radiographic progression-free survival (rPFS) (5.8 months vs. 3.5 months, *p* < 0.001) and a higher objective response rate (ORR) (22% vs. 4%; odds ratio 5.93, 95% CI: 2.01–25.40), compared to the control group receiving enzalutamide or AA. However, with olaparib, there were more grade ≥ 3 adverse events (AEs) [63]. Thanks to these results, the U.S. Food and Drug Administration (FDA) and now the European Medicines Agency (EMA) have approved the use of this drug in this clinical setting.

Rucaparib also received authorization by the FDA after a single-arm phase II CT (TRITON-2), which showed an objective response rate (ORR) of 43.5% and a PSA response rate of 54.8% in BRCA1/2 mutated mCRPC patients who had progressed after new antiandrogen therapies and chemotherapy (from the third-line setting) [64]. However, this approval is conditional on the results of the phase III study TRITON-3 (NCT02975934), and this drug has not yet been approved by the EMA.

Two more PARPi are actually under evaluation by the FDA: niraparib and talazoparib. A phase II single-arm CT (GALAHAD) proved that niraparib in mCRPC patients that have progressed after paclitaxel and AR-targeted therapy could reach a 41% ORR, a 63% complete response rate (CRR), a median rPFS of 8.2 months, and an OS of 12.6 months in the BRCA1/2 mutant population [65]. Another phase II CT (TALAPRO-1) evaluated Talazoparib in DDR-HRR mutated mCRPC patients that progressed after chemotherapy and demonstrated a general ORR of 29.8% and a 46% ORR in BRCA1/2 mutated patients [66].

Recently, some phase III clinical trials have shown the benefit of the combination of a PARPi with a new antiandrogen in the first-line setting of mCRPC. The PROpel trial [67] demonstrated an improvement in rPFS (HR 0.66 [95% CI 0.54–0.81]) of the combination of olaparib and abiraterone compared to abiraterone alone, irrespective of the HRR mutation status. On the other hand, preliminary results from MAGNITUDE showed the benefit of the combination of Niraparib and Abiraterone, with an improvement in the rPFS (relative risk 0.53, 95% CI: 0.36–0.79, *p* = 0.0014) and a reduction in the risk of disease progression/death (47% vs. 27%) in mCRPC with alterations in genes associated with HRR [68].

More clinical trials are currently ongoing with new combination regimens and PCa settings (available at ClinicalTrials.gov).

### 3.2. PARPi as Radiosensitizers in Prostate Cancer

Radiosensitivity has been described based on several factors, including the inherent radiosensitivity of the tumor, its repair capacity, the reoxygenation process, cell cycle redistribution, tumoral tissue repopulation, tumor immunity, and vascular endothelial damage. The combination of ionizing radiation with radio-enhancing agents presents an opportunity to enhance the effectiveness of radiotherapy while minimizing potential damage to healthy tissues and reducing toxic side effects. PARP inhibitors (PARPi) possess several qualities that make them suitable for exerting radiosensitizing effects [69].

#### 3.2.1. The Rationale to Combine PARPi and Radiotherapy

The primary mechanism through which radiation therapy (RT) induces cell death is by causing various types of DNA damage, with double-strand breaks (DSBs) being the most harmful. Therefore, the sensitivity of tumors to ionizing radiation (IR) is closely related to their capacity to repair DSBs [70]. These breaks are repaired through NHEJ and HR [71]. The first one operates throughout all cell cycles, although under high DNA damage levels it does not work properly. HR, however, could occur only in S and G2 phases, as it is the only moment with a sister chromatid available [72]. Following the occurrence of double-strand breaks (DSBs), a rapid phosphorylation of H2AX at S139 takes place, resulting in the formation of γH2AX. This modification serves as a marker for the chromatin surrounding the DSB site and plays a crucial role in facilitating the recruitment of various DNA damage response factors, including 53BP1 [73,74,75]. In PCa, the detection of γH2AX foci has been utilized as a predictive indicator for radiosensitivity [76,77].

Targeting one of these DSB repair pathways can greatly enhance radiosensitivity. However, the challenge lies in minimizing the cytotoxic effect on normal cells while specifically targeting the tumor. BRCA2-deficient cells have a defective HR, so they are forced to activate other DNA repair pathways, such as BER, that are responsible for high-fidelity DDR. In this regard, inhibition of PARP1, which is responsible for BER, may radiosensitize HR-deficient cells. In addition, PARP1 is involved in BER to repair SSBs, and although IR induces numerous SSBs, their efficient repair usually does not lead to substantial cell death. However, PARP1 inhibition, leaves SSBs unrepaired and, together with replication forks, could generate one-ended DSBs. These specific DSBs are predominantly repaired through HR. As a result, PARP inhibition not only exhibits toxicity as a monotherapy but also enhances the radiosensitivity of HR-deficient cells [7,78,79]. Therefore, the combination of PARPi and radiation emerges as a possible treatment based on the ability of PARPi to amplify unrepaired DNA damage [80].

This ability of PARPi to enhance radiosensitivity has been extensively studied in tumors with BRCA1/2 deficiencies, particularly in hereditary breast and ovarian cancers [81]. However, unlike other tumor types, PCa genomes rarely contain BRCA1/2 mutations. Additionally, there is not enough information to date to consider cellular or molecular profiling to personalize treatment with PARPi in PCa. Hence, it is crucial to explore other more prevalent genetic abnormalities that render PCa susceptible to the radiosensitizing effects of PARP inhibitors. One of these mechanisms is the erroneous alternative end-joining (Alt-EJ) pathway, where PARP-1 was found to be crucial. In this context, PARPi can intensify cell death by suppressing homologous recombination and promoting error-prone alt-NHEJ. As a consequence, PARPi might also radiosensitize tumor cells with non-HR deficiencies by suppressing PARP-1-dependent c-NHEJ [82].

This evidence supports the idea that PARPi are potent radiosensitizers, and their combination with radiotherapy may increase oncological outcomes thanks to their probable synergic effect. This combination has been proven effective in other tumors such as breast cancer [9], colorectal cancer [10], pancreatic cancer [11], lung cancer [12], and head and neck cancer [83]. In lung cancer, for example, a phase I CT for the combination of Olaparib + RT with or without cisplatin showed that the maximum tolerated dose (MTD) of Olaparib with RT was 25 mg/24 h, markedly lower than had been anticipated, which emphasizes the potent radiosensitizing properties of olaparib [12]. Regarding breast cancer, a phase I CT suggested that PARP inhibition with olaparib concurrently with radiotherapy for early-stage, high-risk triple-negative breast cancer is well tolerated with no late treatment-related grade 3 or greater toxic adverse events. Three-year overall survival (OS) and event-free survival (EFS) were 83% (95% CI, 70–100%) and 65% (95% CI, 48–88%), respectively. Homologous recombination status was not associated with OS or EFS [9].

#### 3.2.2. Mechanism of Radiosensitization of PARPi

There are different mechanisms described by which PARPi can enhance radiosensitivity in tumors (Figure 3) [84]:

Inhibition of DNA repair: when PARPi are combined with RT, the reparation of SSB is compromised, which leads the cell to DNA replication fork collapse and the appearance of DSB that cause cell death. In addition, PARPi also induce “mechanical” replication fork collapse and consequently DSB. This effect is more potent in BRCA-mutated or BRCAness cells that have a deficient HR. This is an example of a synthetic lethality mechanism [85].

G2/M arrest: when DNA damage occurs, common cells activate checkpoints that lead to cell cycle arrest [86]. PARPi have the capacity to arrest cells in the part of the cycle when they are more sensitive to the radiotherapy effect: in the G2 and M phases [87]. That mechanism enhances RT by maintaining the cell longer in the most radiosensitive phases.

Modulation of chromatin remodeling: PARP-1 inhibition could delay DNA double strand opening and therefore DNA repair [88], favoring the DNA damage caused by the RT.

Replication-dependent radiosensitization: PARPi show their radiosensitizing effect mostly during the cell cycle phase S [89]. Tumors have a higher proliferation rate than the surrounding tissues and therefore more cells in phase S, which help radiosensitize the tumor while saving the surrounding structures.

Impact on the microenvironment and role of hypoxia: Hypoxia induces radioresistance. PARPi show similarities to nicotinamide, a vasodilator, which could help bypass this hypoxia radioresistance [90].

#### 3.2.3. The Combination of PARPi and Radiotherapy in Prostate Cancer: Preclinical Studies

The combination of RT and PARPi is a promising strategy to enhance DNA damage in tumors. Following this idea, some preclinical studies in prostate cancer have shown that novel agents targeting the DNA repair pathway may help increase the efficacy of irradiation while minimizing potential side effects [91] (Table 2).

Han et al. first proved in 2013 that radiation resistance triggered by ERG overexpression increased the efficiency of DNA repair with an amplified expression of γ-H2AX, which could be reversed via PARP1 inhibition. They demonstrated that Olaparib radiosensitized ERG-positive cells by a factor of 1.52 (±0.03) in comparison to ERG-negative cells [92].

In 2015, Gani et al. demonstrated in vitro that AZD-2281 (Olaparib) sensitized 22Rv1 cells to radiation, both under normal oxygen conditions (oxia) and in the presence of acute and chronic hypoxia. In addition, they performed an in vivo study where they showed that combining AZD-2281 with fractionated RT led to a significant delay in tumor growth and increased clonogenic cell death without increasing gut toxicity [93].

Mansour et al. (2017) proved that PTEN plays a role in the repair of DNA double-strand breaks (DSBs) through homologous recombination (HR), as evidenced by increased sensitivity to Olaparib. Their findings showed that while the loss of PTEN is associated with a poorer prognosis in PCa, it may actually indicate a better response to radiotherapy. Additionally, they presented evidence suggesting that PTEN can serve as a biomarker for predicting the response to PARPi as radiosensitizing agents. These findings collectively suggest that PTEN is involved in maintaining genomic stability by delaying the progression of damaged cells into the G2/M phase, thereby providing time for HR-mediated repair of DSBs. Moreover, they identified the PTEN status in PCa as a potential predictor of both radiotherapy and PARPi response, alone or in combination [94].

In 2018, Van de Ven et al. showed that cells resistant to irradiation and tumors derived from a PTEN/Trp53-deficient mouse model of advanced PCa exhibited increased sensitivity to radiation after being treated with NanoOlaparib, a lipid-based injectable nanoformulation of Olaparib. This radiosensitivity was accompanied by changes in the expression of γ-H2AX, which were dependent on the radiation dose and specific to NanoOlaparib. In animals, the combination of NanoOlaparib and radiation tripled the median mouse overall survival (OS) when compared with RT alone, and up to 50% of mice achieved a complete response after 13 weeks [95].

In the same year, Oing et al. reported that BCL2 inhibited the NHEJ repair of DSBs by sequestering the KU80 protein outside the nucleus. They also found that this effect is linked to a shift in DNA repair mechanisms towards error-prone PARP1-dependent end-joining (PARP1-EJ). To support this, they provided in vitro evidence that targeting this repair switch using a PARPi (Olaparib) could selectively enhance the radiosensitivity of cells overexpressing BCL2, offering a promising therapeutic approach. They also corroborated these findings by evaluating retrospectively the impact of BCL2 expression on the clinical outcomes of patients who had been given RT after radical prostatectomy (RP) [96].

With this background, Köcher et al. introduced a functional assay in freshly collected tumor samples from PCa patients that enables the identification of the repair switch to the alternative PARP1-EJ pathway. They demonstrated that an ex vivo assay could be used to detect radiosensitivity in tumor biopsies, helping to personalize treatments [97].

Most recently, Fan et al. demonstrated in LNCaP cells that loss of RB1 enhanced RT DNA damage, inhibiting cell proliferation and provoking cellular senescence through a TP53-dependent pathway. However, when TP53 and RB1 are both deleted, cell proliferation is increased, which facilitates the appearance of castration resistance and RT resistance. Nevertheless, when combined with a PARP1 inhibitor, radiosensitivity was restored [98].

To sum up, all these preclinical trials have shown that the use of PARPi blocking the DDR pathway in combination with RT enhances tumor cell death, as they are not able to repair the damaged DNA caused by RT.

#### 3.2.4. The Combination of PARPi and Radiotherapy in Prostate Cancer: Clinical Studies

Different randomized clinical trials have proven oncological benefits from the combination of RT and ADT in high-risk and locally advanced PC [99,100]. However, even with this treatment, approximately 50% of them will experience biochemical recurrence [101], indicating that better therapeutic regimens are needed. On the other hand, different studies have shown benefits when combining PARPi and ADT, as PARP-1 inhibition suppresses the growth of AR-positive PCa cells. Thus, targeting PARP-1 in PCa seems promising, given that both DNA repair and AR-mediated transcription depend on PARP-1 function [102]. Finally, as shown before, PARPi have exhibited their capacity to radiosensitize tumors in PCa preclinical studies [92,93,94,95,96,97,98]. Together, this opens the research question of combining PARPi, RT, and ADT as a triple therapy.

To clinically establish the potential synergy between PARPi, RT, and ADT, an ongoing phase II randomized CT known as NADIR (NCT04037254) [103] is currently investigating this approach (Table 3). In this trial, 170–180 men with localized high-risk PCa will be enrolled. All patients will receive DE-IMRT and 24 months of ADT and will be randomized to receive or not niraparib for 12 months. The primary endpoint is the proportion of patients with a PSA under 0.1ng/mL after the end of the treatment. The results of this trial, which are still pending, could potentially open up new horizons for the treatment of high-risk PCa.

## 4. Discussion

Radiotherapy is a key treatment for PCa that has traditionally been used for the localized and locally advanced stages [99,100]. However, more evidence is emerging regarding the treatment of the primary tumor in newly diagnosed metastatic PCa. In fact, a secondary analysis of the STAMPEDE trial showed a benefit in OS when treating the primary tumor with RT in patients with less than three bone metastases [104] or with M1a disease [105]. Moreover, two phase II randomized CT studies investigated the role of RT as a metastasis-directed therapy (MDT). The first one, STOMP (n = 62), showed a longer ADT-free survival with an MDT than with surveillance [106]. The second one, ORIOLE, demonstrated a lower progression rate within 6 months with MDT vs. surveillance [107]. Recently, there has been a significant improvement in PFS in favor of MDT (HR: 0.44, *p* < 0.001) when combining the results of STOMP and ORIOLE trials [108].

These data show that radiotherapy is becoming more important in different PCa settings over time. However, there are two main difficulties related to RT as a PCa treatment: the first is the radio-resistance, and the second is the related toxicity. In this scenario, and with the aim of overcoming this issue, different combinations of treatment strategies are arising. One of the most popular ones is the combination of PARPi and RT, which holds solid scientific evidence, as shown in this article.

Nonetheless, to date, only results from a few preclinical studies are available to evaluate the impact of combining PARPi and RT in PCa. Regarding these studies, it is important to note that the majority of them were performed using cell lines derived from metastatic tissue of advanced PCa [92,94,95,96,98] or had selected mutations [92,93,95]. That means that these positive results should be further clinically proven in different PCa settings. Another limitation of these studies is the sample size and the only use of Olaparib [92,93,94,95,96,98] among all PARPi. All PARPi act similarly, but each of them has some particularities, such as a different potency for PARP trapping [109], meaning that these positive results with Olaparib may not be extrapolated to different PARPi. And the other way around, it is possible that other PARPi such as talazoparib, rucaparib, or niraparib achieve better outcomes. Again, this hypothesis should be further studied.

NADIR [103] is the only current clinical trial evaluating the combination of PARPi and RT. In addition, it is the only study assessing the impact of triple therapy with ADT, RT, and PARPi. This approach is supported by the theory that ADT can enhance the radiosensitivity of PCa cells by reducing both the hypoxic fraction [110] and the testosterone-induced increase in DNA repair mechanisms [111,112]. In addition, the DNA-PARP repair pathways are closely connected with the androgen receptor signaling pathway, which is the main regulation pathway for tumor growth in PCa and a therapeutic target for ADT [113].

This clinical trial [103] opens the door for a new horizon in PCa treatment. Actually, there is enough preclinical evidence that encourages the use of PARPi and RT [92,93,94,95,96,97,98]; abundant clinical evidence that supports the positive effect of combining PARPi and ADT [63,64,65,66] as well as PARPi and new antiandrogens [67,68]; and solid evidence for the use of RT and ADT [99,100]. So maybe it is time to explore the combination of the three therapies. Currently, NADIR is evaluating this combination in high-risk localized and locally advanced PCa [103], but perhaps this rationale could also be used in high-risk biochemical failure after a treatment with a curative intention, in low-volume metastatic hormone-sensitive PCa (mHSPC), or even in oligoprogressions in mCRPC.

In fact, emerging data in mCRPC demonstrate that at this stage, the HR defects render these tumors sensitive to PARP inhibition. It seems that there is a dependency on the androgen receptor (AR) to maintain HR gene expression and activity. In addition, after ADT, PARP-mediated repair pathways are upregulated as a mechanism for tumor cell survival, which makes them more sensitive to PARPi. Asim et al. demonstrated in vivo a synthetic lethality between ADT and PARPi, suggesting that ADT may functionally impair HR before the appearance of castration resistance. This finding could potentially be used clinically in advanced or high-risk PCa [114].

To sum up, the combination between PARPi and RT could potentially radiosensitize PCa cells, achieving better oncological outcomes while minimizing undesirable toxicities. However, this combination should be further studied in phase II and phase III clinical trials. In addition, incipient evidence supports the rationale to explore triple combinations with PARPi, RT, and ADT. Nevertheless, this new combination therapy for PCa will have to face the risk of increasing the percentage of severe adverse events, which may be one of the most important limitations, making a well-designed phase I CT essential to determining the MTD.

## 5. Conclusions

RT induces cell death by causing various types of DNA damage, while PARPi inhibit the DNA repair pathway. This rationale makes PARPi a potent radiosensitizer, which has been demonstrated in different tumors. Currently, some preclinical trials have demonstrated positive results with RT and Olaparib in PCa, and an ongoing phase II clinical trial is evaluating the combination of RT, ADT, and niraparib in high-risk and locally advanced PCa. Nevertheless, more randomized clinical trials are necessary to prove the value of this combination with different PARPi and different PCa settings.

## Figures and Tables

**Figure 1 ijms-24-12978-f001:**
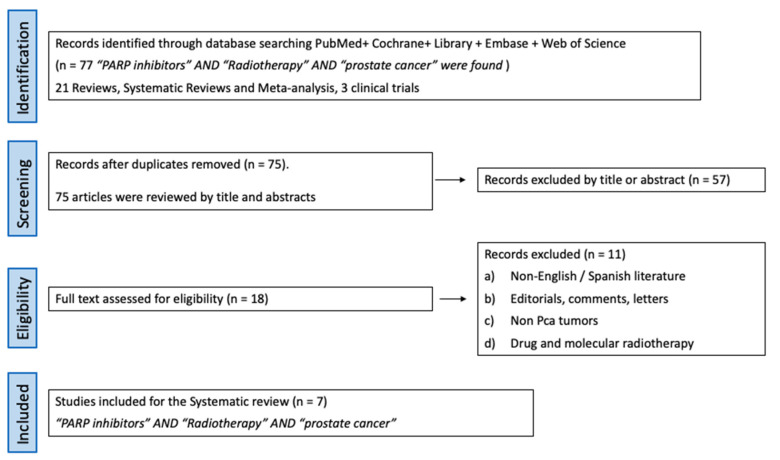
PRISMA flow diagram from the screening process.

**Figure 2 ijms-24-12978-f002:**
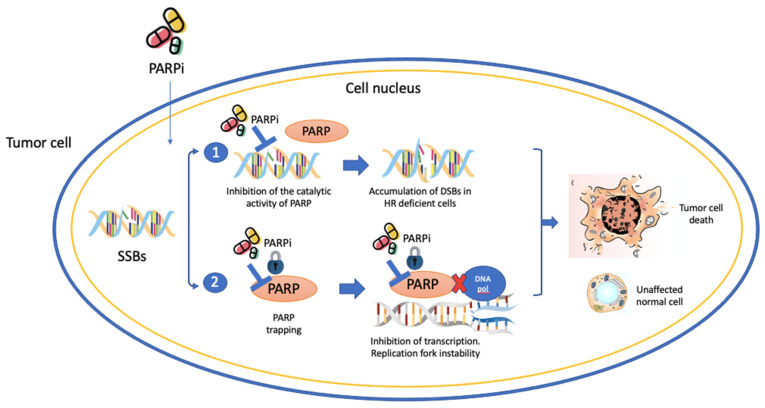
PARPi mechanism of action in the DNA repair process, the transcription and the replication process.

**Figure 3 ijms-24-12978-f003:**
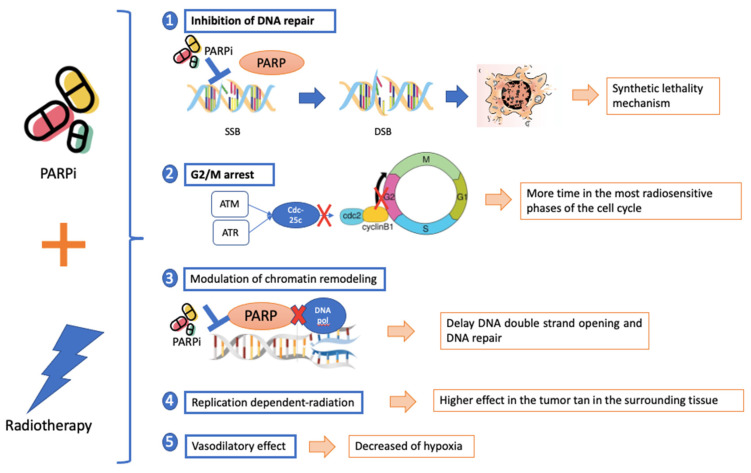
Mechanism of radiosensitization of PARPi that acts in the DNA reparation process in the replication process and in the transcription process.

**Table 1 ijms-24-12978-t001:** Current PARPi with positive results in clinical trials.

Clinical Trial	Drug/Approval Situation	Setting	HRR Mutations	Outcomes
PROfound phase III, randomized [63]	Olaparib vs. AA/Enzalutamide Approved by FDA and EMA	mCRPC who progress after AA or Enzalutamide	Selected	Longer rPFS (5.8 months vs. 3.5 months, *p* < 0.001) Higher ORR (22% vs. 4%; odds ratio 5.93, 95% CI: 2.01–25.40)
TRITON-2, phase II single arm [64]	Rucaparib Approved by FDA	mCRPC who progress after 1–2 novel antiandrogens and paclitaxel	Selected	ORR of 43.5% and a PSA response rate of 54.8%
GALAHAD, phase II single arm [65]	Niraparib Under evaluation by FDA	mCRPC who progress after paclitaxel and AR targeted therapy	Selected	ORR of 41%, CRR of 63%, median rPFS and OS of 8.2 and 12.6 respectively
TALAPRO-1, phase II single arm [66]	Talazoparib Under evaluation by FDA	mCRPC after paclitaxel therapy	Selected	ORR of 29.8%, and 46% BRCA1/2 mutations
PROpel, phase III randomized [67]	Olaparib + AA vs. placebo + AA	First line mCRPC	Unselected	Improvement in rPFS (HR 0.66 [95% CI 0.54–0.81]) in Olaparib + AA arm
MAGNITUDE, phase III randomized [68]	Niraparib + AA vs. placebo + AA	First line mCRPC	Selected	Preliminary results: Improved on the rPFS (relative risk 0.53, 95% CI: 0.36–0.79, *p* = 0.0014) and a reduced the risk of disease progression/death (47% vs. 27%)

**Table 2 ijms-24-12978-t002:** Preclinical studies PARPi + RT.

Author, Year	In Vitro: Cell Line and Intervention	In Vivo: Animal Model and Intervention	Ex Vivo: Model	Outcomes
Han et al., 2013 [92]	PC3 and DU145 PCa cell lines with ERG overexpression Intervention: 1.0 μM Olaparib 1h pre Radiation (2 Gy/min)	Xenograft models: subcutaneous mice model of PC3-control and PC3-ERG Intervention: ABT-888 (100 mg/kg) twice daily, or radiation alone (2 Gy for 5 days), or in combination		In vitro: olaparib radiosensitized ERG-positive cells by a factor of 1.52 (±0.03) relative to ERG-negative cells In vivo: ERG overexpression confers radiation resistance through increased efficiency of DNA repair following radiation that can be reversed through inhibition of PARP1.
Gani et al., 2015 [93]	22Rv1 PCa cells Intervention: RT 2 Gys + 1.0 μM PARP inhibitor (AZD-2281)	22Rv1 prostate xenografts: Old male CD1 nude mice were injected s.c. with 2.0 × 10^6^ cells Intervention: (a) vehicle alone; (b) AZD-2281 100 mg/kg. daily for 3 days; (c) 8 Gy plus vehicle i.p. for 3 days; (d) 8 Gy plus AZD-2281 100 mg/kg for 3 days; (e) 5x2 Gy plus 7 days vehicle i.p.; (f) 5x2 Gy plus 7 days of AZD-2281 100 mg/kg		Combining AZD-2281 with fractionated RT led to a significant delay in tumor growth and increased clonogenic cell death without increasing gut toxicity
Mansour et al., 2017 [94]	Cell lines DU145, BPH1 Intervention: Irradiation of 0.8 Gy/min + 1 μM olaparib for 4–6 h		Tissue spots from 3261 radical prostatectomies (RP)	PTEN status in PCa may be a potential predictor of both RT and PARPi response, alone or in combination
Van de Ven et al., 2018 [95]	Three PTEN-deficient PCa cell lines: human PC3 (ATCC), human LNCaP (ATCC), and mouse FKO1 Intervention: 1μM Olaparib or NanoOlaparib for 24 hr, irradiated (0–10 Gy)	Subcutaneous mouse model of radiation resistant PCa generated from FKO1 Intervention: NanoOlaparib (40 mg/kg) IV twice weekly 12 weeks + irradiation one dose of 10 Gy		In vitro: radiosensitivity increased after NanoOlaparib. Changes in γ-H2AX In vivo: Triple the mouse OS compared to RT alone. 50% complete response after 13 weeks. MRI studies revealed that NanoOlaparib enhances the intratumural accumulation of systemically administered nanoparticles (ferumoxytol)
Oing et al., 2018 [96]	Cell lines DU145, BPH1 Intervention: Irradiation of 0.8 Gy/min + 1 μM olaparib		Samples from RP	Olaparib could enhance radiosensitivity of cells overexpressing BCL2
Köcher et al., 2019 [97]			Freshly collected tumor samples from PCa	an ex vivo assay could be used to detect radiosensitivity in tumor biopsies helping to personalize treatments
Fan et al., 2021 [98]	Cell lines: LNCaP, 22RV1, PC3, and DU145 Drugs: Enzalutamide, KU55933, Olaparib (AZD2281) RT: 2–8 Gy			Loss RB1 increased vulnerability to the DNA damage pathway. RT + PARP1 increased radiosensitivity.

**Table 3 ijms-24-12978-t003:** Clinical Trials PARPi + RT.

Study	Design	Estimated Enrollment	Setting	Intervention	Primary Endpoint
NADIR (NCT04037254) [103]	Phase II, randomized clinical trial	180	High risk localized or locally advanced PCa (no prior treatment) with or without HRR mutations	Niraparib + RT + ADT vs. niraparib alone vs. RT + ADT	Maintenance of disease-free state

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
