# Peer review of "Parp Inhibitors and Radiotherapy: A New Combination for Prostate Cancer (Systematic Review)"

_ijms, 2023, doi:10.3390/ijms241612978_

Round 1
Reviewer 1 Report
This is a very well-written and informative review, adequately summarizes ongoing progress and clinical trials at various stages for PARPi and RT, and also presents the case for triple therapy (ADT+PARPi + RT).
Author Response
Thank you very much for your kind review report.
Reviewer 2 Report
Comments for authors
Overall, the manuscript is well-organized and addresses an interesting and significant subject. However, prior to publication, substantial revisions are necessary. Authors must pay attention to each comment and revise the manuscript accordingly.
Comment 1: First of all, this manuscript is a full review article, which means authors need to discuss the subject in detail for new readers, especially background and mechanisms. Only 77 references were provided which shows a poor literature review. This review requires a minimum of 100 references, incorporating the most recent advancements in the field.
Comment 2: The introduction needs substantial improvements. For new readers, the author's background may not be sufficient to understand the significance of this study. It's also unclear how the radiations affect the cells or biological systems. In a review article, I recommend that authors include the suggested article in the introduction section to strengthen the background and mechanisms by which the radiations interact and affect the biological systems.
Article: Microwave Radiation and the Brain: Mechanisms, Current Status, and Future Prospects. International Journal of Molecular Sciences vol. 23 (2022). [https://doi.org/10.3390/ijms23169288].
Comment 3: Can you elaborate on the three primary mechanisms for DNA repair initiated by cells in response to ionizing radiation, namely non-homologous end-joining (NHEJ), homologous recombination (HR), and base excision repair (BER), and how do PARPi sensitize cells by leveraging the BER pathway?
Comment 4: How does the combination of PARPi and radiotherapy heighten the likelihood of replication forks collapsing and lead to the formation of persistent double-stranded DNA breaks (DSBs)? Explain in the manuscript.
Comment 5: What are the key factors contributing to the potency of the PARPi and radiotherapy combination as an oncological strategy, and how have these factors been demonstrated in various tumor types? Explain in the manuscript with details.
Comment 6: What evidence exists from preclinical studies to support the combination of PARPi and radiotherapy in the context of prostate cancer (PCa), and what are the significant findings from the ongoing clinical trial in this specific cancer type? It is important if authors can provide the reference of recent studies covering these questions if available.
Comment 7: What are the potential challenges and limitations associated with the use of PARPi and radiotherapy as a combined treatment approach in prostate cancer, and how can these be addressed in future research? Discuss them in the conclusion section for clear understanding.
Comment 8: Can you provide insights into the molecular and cellular processes underlying the synergy between PARPi and ionizing radiation, specifically in the context of PCa, and how these findings compare to other tumor types?
Comment 9. Considering the complexity of DNA repair pathways and tumor heterogeneity, how can advancements in personalized medicine and molecular profiling contribute to optimizing the selection of patients most likely to benefit from PARPi and radiotherapy combination therapy in PCa?
Comment 10: I encouraged the authors to provide detailed figure legends for Fig. 2 and Fig. 3, aiming to enhance reader feasibility and facilitate better understanding.
Comment 11: There are typos and inaccuracies in the paper. I strongly recommend authors read precisely and correct the grammatical errors.
There are typos and inaccuracies in the paper. I strongly recommend authors read precisely and correct the grammatical errors.
Author Response
Thank you very much for the paper revision. The article has been reviewed according to your recommendations and all the comments have been fully addressed:
- Comment 1: The paper has now 114 references.
- Comment 2: The introduction has been modified and the mechanism of action of ionizing therapy has been detailed for a better comprehension. The recommended article has been also included. In addition the aim of the revision and the importance of the study has been explained in detailed.
- Comment 3: the 3 mechanism for DNA repair have been extensively explained in the introduction.
- Comment 4: the replication fork has now been explained with more detail in the introduction
- Comment 5: The actual evidence regarding other tumor types have been explained in the epigraph "3.2.1. The rationale to combine PARPi and radiotherapy" (in red)
- Comment 6: The evidence from preclinical studies to support the combination of PARPi and RT have been explained in the parapraph: “3.3.3 The combination of PARPi and Radiotherapy in prostate cancer: preclinical studies” and is now sum up at the end. There are no available results from the only CT (in red).
- Comment 7: the most important limitation for the combination and how it could be addressed has been detailed at the end of the discussion (line 446-449)
- Comment 8: the processes underlying the synergy between PARPi and ionizing radiation are explained in the paragraph 3.2.1 and the introduction (in red the extended information for the revision)
- Comment 9: there are not yet enough information to have a molecular profiling contribute to optimizing the selection of patients most likely to benefit from PARPi and radiotherapy combination therapy in PCa. It has been explained in the paragraph 3.2.1 (lines 265-275)
- Comment 10: The figure legend has been explained with more detail.
- Comment 11: The paper has beed revised to correct all grammatical errors.
Reviewer 3 Report
PARPi and RT relation is an actual concern for many scientists.
Thank you for providing this interesting review. The paper is clear and well-written. authors followed PICOS guidelines. Inclusion and exclusion criteria are correctly explained.
Please add just a line or two on the effect of PARP-1 Inhibitors on the transcriptional activity of androgen receptor PCa cells.
For example:
https://doi.org/10.1007/978-1-0716-2891-1_19
Author Response
Thank you very much for the paper revision. The article has been reviewed according to your recommendation regarding the effect of PARP-1 Inhibitors on the transcriptional activity of androgen receptor PCa cells.The reference and the explanation has been added in lines 379-381
Round 2
Reviewer 2 Report
The authors have addressed all of my comments and concerns in the revised version. I recommend accepting the article in present form.